# Microseismic Monitoring Signal Waveform Recognition and Classification: Review of Contemporary Techniques

Hongmei Shu 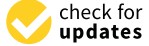 and Ahmad Yahya Dawod *

International College of Digital Innovation, Chiang Mai University, Chiang Mai 50200, Thailand;
hongmei_shu@cmu.ac.th
* Correspondence: ahmadyahyadawod.a@cmu.ac.th

**Abstract:** Microseismic event identification is of great significance for enhancing our understanding of underground phenomena and ensuring geological safety. This paper employs a literature review approach to summarize the research progress on microseismic signal identification methods and techniques over the past decade. The advantages and limitations of commonly used identification methods are systematically analyzed and summarized. Extensive discussions have been conducted on cutting-edge machine learning models, such as convolutional neural networks (CNNs), and their applications in waveform image processing. These models exhibit the ability to automatically extract relevant features and achieve precise event classification, surpassing traditional methods. Building upon existing research, a comprehensive analysis of the strengths, weaknesses, opportunities, and threats (SWOT) of deep learning in microseismic event analysis is presented. While emphasizing the potential of deep learning techniques in microseismic event waveform image recognition and classification, we also acknowledge the future challenges associated with data availability, resource requirements, and specialized knowledge. As machine learning continues to advance, the integration of deep learning with microseismic analysis holds promise for advancing the monitoring and early warning of geological engineering disasters.

**Keywords:** microseismic events; machine learning; signal processing; waveform recognition; image classification; sensor technology

## 1. Introduction

In the contemporary landscape of mining operations, a profound transformation is underway, marked by the transition to intelligent mining [1]. This paradigm shift is catalyzed by the integration of cutting-edge technologies such as the Internet of Things (IoT) [2], big data analytics [3], and artificial intelligence (AI) [4]. Central to this transformation is the unwavering commitment to enhancing mining safety [5]. As mining operations plunge deeper into the Earth's crust, the stability of rock formations is increasingly susceptible to disruptions caused by human activities [6]. These disruptions give rise to geological hazards like rock bursts and mining-induced seismic events, posing grave threats to the safety of miners and the productivity of mining endeavors [7,8]. Within this context, microseismic monitoring technology has emerged as a fundamental pillar for ensuring geological safety in mining operations (Figure 1).

Microseismic monitoring entails the continuous surveillance of minuscule seismic events during mining activities [9]. These imperceptible events provide valuable information about evolving geological conditions. They serve as early warning signals, offering crucial insights into potential hazards and enabling timely preventive measures. This not only safeguards the well-being of miners but also enhances the overall efficiency and sustainability of mining practices [10]. In addition to its practical value and significance in smart mining, microseismic monitoring technology is widely used in various fields. By searching the Web of Science database using the keyword "microseismic monitoring", we

selected a total of 1286 articles published between 2012 and 2023, spanning the past decade. Based on the titles, abstracts, and keywords of these articles, we classified them into four application domains: underground energy exploration and development (e.g., oil and gas resources), real-time monitoring, and early warning of geological hazards (e.g., landslides and karst collapses), seismic activity research and earthquake monitoring, and monitoring and evaluation of underground storage and waste disposal facilities (e.g., nuclear waste repositories). The statistical results are shown in Figure 2.

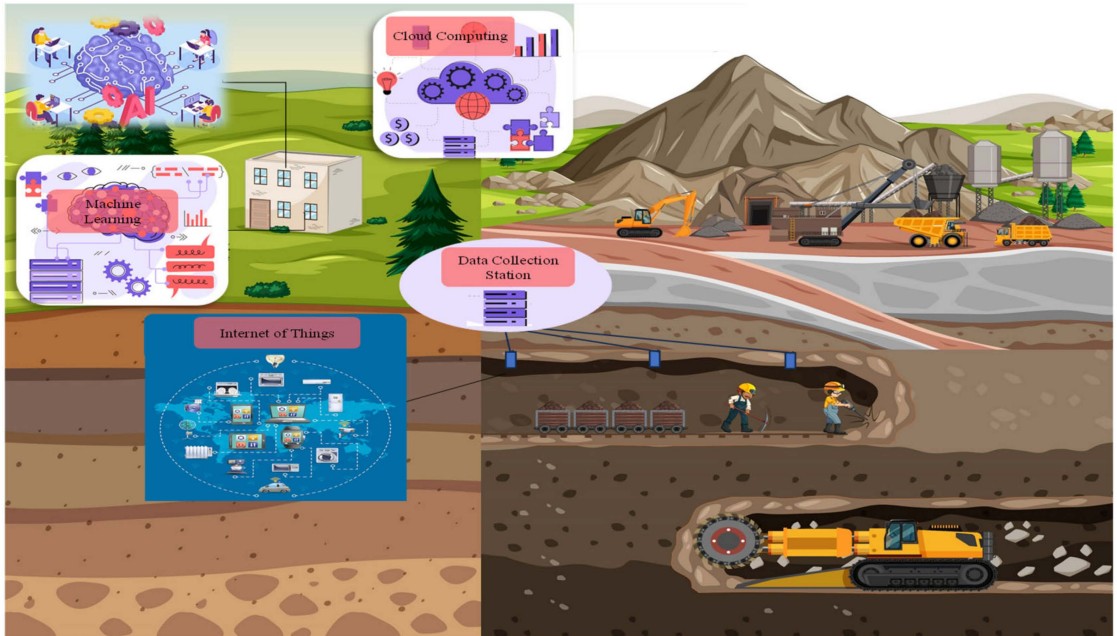

**Figure 1.** Application of microseismic monitoring in smart mining.

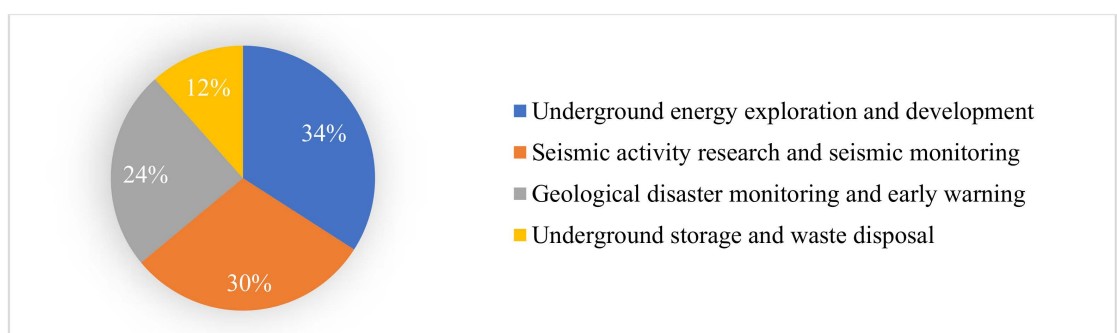

**Figure 2.** Application of microseismic monitoring technology in various fields.

Considering the changing mining environment and the need for safety assurance, we initiated this review of contemporary techniques applied to the recognition and classification of microseismic events. This work not only addresses the pressing need for advanced methods, but also aligns with the broader trend of intelligent mining practices. By combining deep learning with microseismic event analysis, we aim to provide a comprehensive survey for the geological engineering and computer science communities, enabling them to harness the potential of AI-driven microseismic monitoring.

The fracturing of rock masses during seismic events, geological disasters, and underground excavation processes generates abundant microseismic signals (MS) [11]. Microseismic monitoring systems (MMS) used in deep subterranean engineering can collect a vast amount of waveform data in real time [12]. Certain deceptive noise signals, on the other hand, bear striking similarities to microfracture signals, requiring engineers to cross-reference signal features from distinct domains [13]. This challenge poses difficulties in

rapidly and accurately detecting microseismic data. Notably, the presence of noise signals closely resembling microseismic signals significantly hampers swift and precise detection, consequently undermining effective geological hazard assessment and risk prevention [14].

MMS generates vast volumes of data, primarily capturing microseismic events—a task fraught with challenges due to the intricate nature of on-site monitoring environments. Traditionally, the identification of these events relied heavily on manual experience. Subsequently, statistical analysis and spectral analysis methods were employed in an attempt to decipher the complex signals embedded in the data. However, these methods often fell short of effectively separating valuable MS from the sea of data that includes numerous irrelevant noise signals [15].

The complexities of the monitoring environment and the dynamic nature of the collected signals present significant challenges. These traditional approaches require operators to possess a deep understanding of geophysics and signal processing to establish precise identification criteria. However, due to the inherent variability in signals, achieving consistent and accurate identification has proven to be a formidable task. Additionally, traditional methods of identifying microseismic events have become time-consuming and inefficient as monitoring datasets continue to grow [16]. Nonetheless, the rise of machine learning, particularly deep learning approaches, has shown remarkable accuracy in recognizing and classifying microseismic occurrences. Deep learning algorithms, when compared to standard methods, have overcome constraints and offer intriguing possibilities for complicated monitoring situations [17].

While various studies have examined the use of machine learning in MMS, there has yet to be a comprehensive study of the combination of deep learning and image recognition in this context. We undertook this study to fill this void and conduct targeted research. Our goal is to explore the complexities of using deep learning to microseismic event waveform picture detection and classification, providing insights into its potential and limitations. The significance of this work is that it focuses on the investigation of cutting-edge deep learning applications in microseismic monitoring. We provide a complete overview of the available methods and a comparative study of their strengths and limitations through an in-depth examination of picture recognition. Furthermore, this study serves as a vital reference for researchers, practitioners, and stakeholders in geological engineering and earth sciences. It lays the groundwork for future advancements in geological hazard prediction and monitoring, ultimately enhancing safety in mining operations. As deep learning continues to advance, its seamless integration into microseismic analysis promises valuable insights and practical applications, bolstering geological safety in mining activities.

The rest of the article is organized as follows: Section 2 presents the materials and methods and provides detailed descriptions of the microseismic monitoring system, microseismic monitoring data, and waveform characteristics. We also summarize the representative literature on microseismic event identification and outline various identification methods. Section 3 focuses on the analysis and discussion of the results. We classify existing methods, conduct a SWOT analysis of deep learning methods, compare existing models, and explore future development opportunities and challenges. Finally, we conclude this study by summarizing the findings and proposing possible directions for future research.

## 2. Materials and Methods

### 2.1. Microseismic Monitoring Signals

Microseismic events refer to weak seismic activities caused by minor displacements and stress changes within underground rock formations [18]. In mining environments, factors such as mining activities and rock movements can lead to minor fractures and deformations in rocks, resulting in microseismic events. These events can occur naturally or be triggered by human activities like mining operations and blasting. Monitoring and analyzing microseismic events are crucial for mine safety. Using the example of MMS in a metal mine, the system collects underground microseismic signals. These signals include seismic waveforms from microseismic events, which record the seismic signals generated

when underground rock formations undergo slight changes. These signals may encompass different types of events such as rock fractures and blasting. The monitoring system captures these signals using sensors and transmits them to the surface or data centers for further analysis. These signals exhibit unique characteristics in terms of waveform, amplitude, frequency, etc., which can be used to identify various types of microseismic events and provide essential information and warnings for mine safety [19].

For instance, microseismic monitoring systems play a crucial role in ensuring coal mine safety by capturing MS from coal-rock fractures and blasting activities. However, the challenge lies in distinguishing between these signals due to their waveform similarities. Scholars have extensively studied this issue to enable accurate identification of authentic MS within the monitoring system [20,21]. Seismic data analysis methods, commonly used for assessing seismic activity in volatile mines, encounter challenges due to localization errors and incomplete data catalogs caused by unfavorable seismic detector layouts [22]. Furthermore, dynamic disasters like stress-type and fracture-type rock bursts significantly impact mine safety. To address this, a study integrates spatiotemporal parameters through a big data platform and employs the AdaBoost algorithm to predict rock burst risks, contributing to accurate and timely warnings [23]. Shu et al. examined the features and classification of MS in coal mine workings, as well as its importance in the early detection of gas and coal outbursts [24]. In addition, Yin et al. employed a data-driven approach based on deep learning to successfully predict coal seam floor water inrush using microseismic monitoring data [25]. These applications highlight the significance of MMS and data analysis in enhancing safety production in coal mining.

### 2.1.1. Microseismic Monitoring Data

Microseismic monitoring data exhibit unique characteristics that are vital for effective event recognition and classification [26]. Firstly, these data demonstrate variations in signal intensity over both time and space, which may be attributed to complex changes in underground rock formations and mining activities. Second, mechanical processes, equipment breakdowns, subsurface water movement, or environmental conditions can contribute to high levels of noise in microseismic data. This complicates data processing because a good distinction between signals and noise is required. Furthermore, due to geological effects, the waveforms of microseismic data are frequently complex and diverse, reflecting the physical qualities of rocks and differences in subsurface structures. Finally, microseismic data span a wide range of frequencies and energy levels, implying that MS can manifest in a variety of frequency ranges and energy levels. This variability causes difficulties in event recognition and classification, necessitating the investigation and processing of various features.

These microseismic surveillance data characteristics represent the complexity of the underlying rocks and mining activities, which require advanced data analysis techniques for interpretation and understanding. Variations in signal intensity across time and place suggest that the distribution of microseismic events is nonuniform, which may be related to the nonuniformity of underground rock layers or mining activity. To improve the detection and analysis of microseismic events in the presence of high noise levels, preprocessing measures such as blurring and filtering are required. The complex waveforms may reflect a variety of subsurface structures, which is important for recognizing various types of occurrences such as rock bursts, demolition, and microfractures. Because of the variety of frequencies and energy levels, multiscale and multifeature analysis approaches must be used to capture the many properties of microseismic signals.

Understanding these properties of microseismic monitoring data is therefore critical for accurate identification and classification of microseismic occurrences. When confronted with such complicated data, researchers must devise proper data processing and analysis methods to distinguish between important events and irrelevant noise, as well as extract key information about microseismic events. Furthermore, different types of microseismic events may necessitate alternative feature extraction and classification approaches to ensure

accuracy and reliability. These data features enable in-depth investigations of subterranean engineering and rock behavior and provide critical information for mining activities and underground engineering projects.

### 2.1.2. Waveform Features

Microseismic events manifest in various forms of waveform images, each possessing unique characteristics. To comprehensively analyze and classify microseismic events, researchers can employ methods for extracting relevant features from these waveform images. Figure 3 illustrates different types of microseismic event waveform images, encompassing microseismic events (commonly referred to as rock microfracture events), blasting events, rock drilling events, power interference events, and other noise events. The diversity in these images reflects the waveform features of different events; thus, feature extraction from these images can aid in better understanding and distinguishing various types of microseismic events.

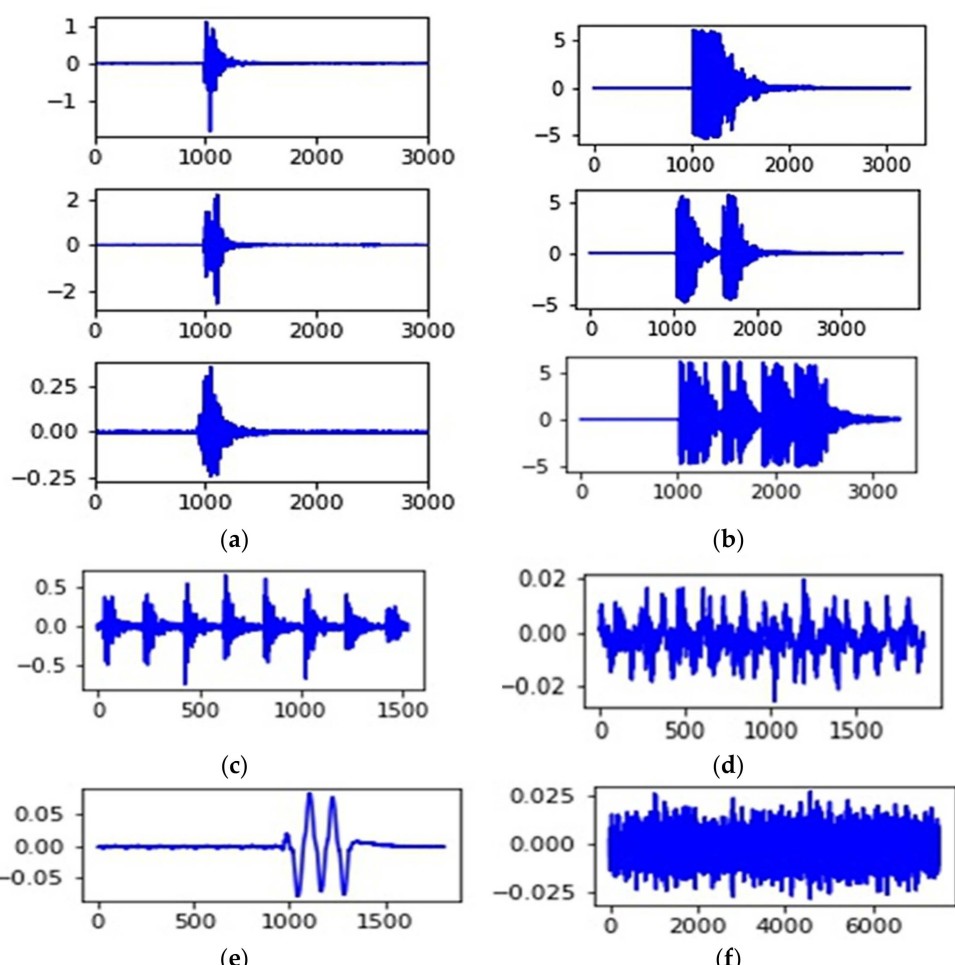

**Figure 3.** Various types of microseismic event waveform images. (**a**) Different waveforms of rock microfracture (**b**) Different waveforms of blasting (**c**) Drilling (**d**) Orepass blasting (**e**) Electrical noise (**f**) Background noise.

Microseismic event classification can typically be based on triggering mechanisms, waveform characteristics, and occurrence locations [27]. Common classifications of microseismic events include naturally occurring microseismic events, which are triggered by natural processes such as underground structural movement and rock deformation; human-induced microseismic events, which are caused by human activities such as mining operations and blasting; and rock burst events, which are caused by the rupture or collapse

of rocks. Based on different triggering mechanisms, monitored microseismic events can be divided into the following categories:

1. Rock microfracture microseismic events, which are usually associated with stress changes and cracks in underground rock formations. The waveform characteristics of these events show small amplitude, high frequency, short duration, and may display certain periodicity.

2. Blasting events are seismic signals generated by blasting activities in mines or underground projects. The waveform characteristics are usually high amplitude, low frequency, longer duration, and specific spectral features.

3. Rock drilling events, which originate from mechanical activities such as rock gouging and drilling in mines and other sites, produce noisy signals. Although these signals may appear in microseismic monitoring, they are unrelated to seismic activity and need to be distinguished from microseismic events. Their waveform characteristics are generally characterized by high amplitude noise, broad bandwidth, transient signals, and lack of significant periodicity.

4. Other noise events, which refer to noise sources other than rock drilling events, such as unloading, equipment operation, power interference, and groundwater flow, may interfere with the monitoring and identification of MS. The waveform characteristics of these noise events are usually random, irregular, and without obvious frequency and amplitude patterns.

Extracting features from waveform images is a critical step in microseismic event analysis because these features can be used to characterize the event in both the time and frequency domains. These features may include amplitude, frequency, energy distribution, waveform shape, etc. By quantitatively analyzing these features, researchers can establish models for microseismic event recognition and classification [28]. These models can assist mining and underground engineering monitoring systems in more accurately identifying and responding to potential microseismic events, thereby enhancing the safety and sustainability of underground operations. Figure 3 provides a visual reference, emphasizing the diversity of waveform images and underscoring the necessity of feature extraction from these images. By subdividing microseismic events based on different classification criteria, we can gain a more accurate understanding of different types of underground activities and their potential impacts, thus better safeguarding the safety of mines and projects.

### 2.2. Literature Summary

This study's resources were obtained from credible academic databases such as Google Scholar, Web of Science, Scopus, and PubMed. The study information was indexed using a preset set of keywords such as "microseismic event", "microseismic waveform", "machine learning", "deep learning", "image recognition", and "image classification". An extensive search of Google Scholar was undertaken and relevant studies were categorized according to subject significance.

A thorough analysis of the existing literature is crucial for understanding methodologies in microseismic event waveform recognition and classification. By examining current research, we can draw insights from predecessors to guide future studies. Table 1 summarizes the important discoveries, limitations, and pros and cons of various methods used in this field. It provides a comprehensive reference for researchers, encompassing statistics, spectral analysis, traditional machine learning, and deep learning approaches. These findings offer valuable insights into the strengths and limitations of different methods of addressing the challenges of microseismic event recognition and classification.

**Table 1.** Summary of existing research.

| Scholars | Method/Objectives | Key Findings | Limitations/Gaps |
|---|---|---|---|
| Lin et al., 2018 [29] | Using a deep convolutional neural network with spatial pyramid pooling (DCNN-SPP) method for joint recognition and classification of multichannel microseismic waveforms. | Automatic updating of network parameters was achieved by directly processing multichannel waveforms through end-to-end training and testing. The classification accuracy of the test set was 91.13%. | Training data are limited, and classification accuracy can be further improved. |
| Binder and Chakraborty, 2019 [30] | Using CNNs to detect microseismic events in a comprehensive distributed acoustic sensing (DAS) strain wavefield. | Neural networks provide low-cost, automated detection of microseismic events. | The study only compared the STA/LTA algorithm, lacking comparisons with other advanced algorithms. |
| Lin et al., 2019 [31] | Using a hybrid technique of DCNN and support vector machine (SVM) to identify and classify multichannel microseismic waveforms. | The DCNN–SVM method outperformed random forests (RF) and k-nearest neighbors (KNN), with an accuracy rate of 98.18%. | Preprocessing of the dataset, including noise reduction and filtering, required a significant amount of time and effort. |
| Zhang et al., 2019 [32] | Combining ensemble empirical mode decomposition (EEMD), singular value decomposition (SVD), and extreme learning machine (ELM) for the automatic identification of microseismic data. | ELM outperformed backpropagation neural networks, neural networks optimized with genetic algorithms, and SVM classification models. It achieved an average recognition accuracy of 93.85% with a training time of 0.15 s. | The data preprocessing process is relatively cumbersome. |
| Dong et al., 2020 [16] | A CNN-based image recognition model for microseismic events and explosions in seismic waveforms. | The CNN-based model achieved 99.46% accuracy for microseismic events and 99.33% accuracy for explosions in the test dataset using original seismic waveform images. | The study lacks stability and robustness testing of the model using actual data from different mines. |
| Kang et al., 2020 [33] | Classified microseismic events and explosions using a deep belief network (DBN). | The model outperformed the accuracy obtained with SVM and Fisher classifiers, achieving 94.4%. | This method still requires combining source data and selecting feature parameters. |
| Peng et al., 2020 [34] | Used capsule network (CapsNet) for the automatic classification of limited sample microseismic records. | On a small number of training examples, the approach obtained 99.2% accuracy. It surpassed CNN and other machine learning algorithms in terms of performance. | There is a lack of robustness testing with actual data from different mining sites. |
| Song et al., 2020 [35] | Identification of mining microseismic and blast signals using CNN and Stockwell transform-based color images. | Leveraged the advantages of CNN in image recognition by directly training on raw images of microseismic signals, thereby avoiding cumbersome data preprocessing. | The sample data are limited, and the model's stability and robustness require further testing. |
| Wei et al., 2020 [36] | Proposed a waveform image discrimination method using principal component analysis (PCA) + SVM for automatic classification of microseismic events and explosions | Combining waveform image features extracted using PCA with the SVM classifier can accurately identify microseismic events with an optimal result of 90%. | Lack of comparison with existing research methods, and classification accuracy still has room for improvement. |
| Yi et al., 2020 [37] | Using CEEMDAN_SE (complete ensemble empirical mode decomposition with adaptive noise sample entropy) for feature extraction and the classification of mining microseismic signals. | Significant differences in sample entropy values exist between microseismic and blast signals. When combined with ELM, CEEMDAN_SE achieves a classification recognition accuracy of 91.5%. | Larger and more diverse training datasets are needed for extensive research to quantify the robustness of this method. |

**Table 1.** *Cont.*

| Scholars | Method/Objectives | Key Findings | Limitations/Gaps |
|---|---|---|---|
| Bi et al., 2021 [38] | Proposed an understandable time–frequency convolutional neural network (XTF-CNN) for microseismic waveform categorization. | XTF-CNN outperforms competing methods (CNN, LSTM, RNN-FCN, and ResNet) in both performance and interpretability. | Larger-scale research is needed to quantify the robustness of this method. |
| Jiang et al., 2021 [39] | Proposed a method for automatic discrimination between microseismic and blasting events based on time–frequency spectral features. | This method reduced the operator's sensitivity to classification, and improved the accuracy and efficiency of mass spectrometry signal data identification in spectral monitoring technology applications. | Larger-scale research is needed to quantify the robustness of this method, including considering the influence of different blasting schemes on the identification results. |
| Peng et al., 2021 [40] | Used the deep convolutional neural network inception (DCNN–inception) algorithm for microseismic data recognition. | The DCNN–inception algorithm outperformed CNN in recognition accuracy. | This method has a long training time and requires a large amount of data to refine the network. |
| Rao et al., 2021 [41] | Employed a particle swarm optimization (PSO) algorithm to optimize the ELM artificial intelligence model, PSO–ELM, to discriminate microseismic events and explosions. | Compared with the original ELM model and other commonly used intelligent discrimination models, PSO–ELM demonstrated the best discrimination performance. | The method still requires the manual selection of feature parameters. Additionally, it was tested using data from only one mining site, and its generalizability requires further exploration. |
| Tang et al., 2021 [42] | Proposed a CNN architecture with an attention mechanism to automatically identify microseismic events. | This model can improve network performance by enhancing the intermediate information in CNN without many additional parameters or significant computational costs. In addition, the multichannel model can achieve the best results. | Further analysis and discussion are needed regarding the selection of the number of channels in the multichannel model. |
| Zhao et al., 2021 [43] | Developed a hybrid model combining singular spectrum analysis (SSA), CNN, and long short-term memory network (LSTM) to identify microseismic signals. | Compared with common identification methods, including CNN, LSTM, backpropagation neural network (BP), SVM, decision tree (DT), KNN, and linear discriminant analysis (LDA), this hybrid model achieved higher recognition accuracy. | The performance of this method was tested using data from only one mining site, so its generalizability requires further exploration. |
| Ding et al., 2022 [44] | Proposed an improved neural network combined with transfer learning for the recognition of mining microseismic events. | The improved T-SimCNN transfer learning model achieved a recognition accuracy of 95% for microseismic events. | Further large-scale research is required to quantify the robustness of this method. |
| Fan et al., 2022 [45] | Proposed a wavelet scattering decomposition (WSD) transform combined with an SVM algorithm for discriminating microseismic signals with low signal-to-noise ratios. | The scattering coefficients of each signal proved to be suitable as features for training unique models. In experimental samples, this model achieved 92.86% recognition accuracy. | The method was compared only with the standard STA/LTA algorithm and lacks comparison with other more advanced algorithms. |
| Jia et al., 2022 [46] | Designed three earthquake event classifiers referencing the CNN structures VGGnet, ResNet, and Inception. These classifiers were tested and compared using three-channel seismic full waveform time series and spectral data. | The test dataset includes three classes of events: natural earthquakes, explosions, and collapses. The results showed that the classifier achieved recall and accuracy rates exceeding 90%. | Lack of comparison with other commonly used research methods. |

**Table 1.** *Cont.*

| Scholars | Method/Objectives | Key Findings | Limitations/Gaps |
|---|---|---|---|
| Li et al., 2022 [15] | Recognition and classification of MS waveform images and spectrograms using deep learning models, including VGG16, ResNet18, AlexNet, and their ensemble models. | Each model performed well on the non-denoised waveform image set, with accuracies of 96% for AlexNet, 98% for VGG16, 96% for ResNet18, and 98% for the ensemble model. | The performance of the method requires further validation on data from multiple mining sites to assess its generalizability. |
| Wang et al., 2022 [47] | Proposed an enhanced convolutional natural network (ECNN) based on the ACGAN structure for MS waveform classification. In this approach, the generator synthesizes samples of specified types, and the discriminator identifies class and authenticity. | The study investigated how changes in training samples affect ECNN and traditional CNN models. The results indicate that the classification accuracy of both models stabilizes when the number of samples exceeds 1024. | Further research on a larger scale is needed to quantify the robustness of this method. |
| Wang et al., 2022 [48] | Proposed a dual-channel CNN model (T-WPD CNN) with time domain information and wavelet packet decomposition coefficients. | Wavelet packet decomposition factors emphasize signal qualities while suppressing noise characteristics. T-WPD CNN surpasses typical CNN approaches in terms of reliability and resilience, according to experimental results. | The time cost of signal processing requires further consideration. |
| Chen et al., 2023 [49] | Conv–LSTM–Unet is a deep learning model which utilizes convolutional neural networks (Conv) and long short-term memory networks (LSTM) for microseismic signal detection. | The Conv–LSTM–Unet model adopts a semantic segmentation method to extract the spatiotemporal aspects of microseismic data more effectively. It is less susceptible to noise and outperforms conventional recognition models. | Future research should focus on addressing the model's generalization performance. |
| Ma et al., 2023 [50] | Deep learning techniques and short-time Fourier transform (STFT) technologies were used to develop an accurate microseismic signal recognition and classification model. | STFT time–frequency analysis exposes distinct properties of noise, microseismic, and blasting signals, allowing for fine time domain distinction from noise signals that approach MS. | Future research should focus on conducting larger-scale studies to quantify the robustness of this method. |
| Dong et al., 2023 [51] | Using CNN-based transfer learning models for microseismic event waveform classification | Four types of microseismic event datasets were constructed, and transfer learning was performed on pretrained AlexNet, GoogLeNet and ResNet50 models. Compared with the SVM classifier, GoogLeNet has the best overall performance with 99.8% recognition accuracy. | Future research should focus on conducting larger-scale studies to quantify the robustness of this method. |

1. Methodological diversity: The table highlights the diversity of methodologies employed in microseismic event recognition, covering a spectrum from traditional methods, such as EEMD (Table A1 shows the full names of all abbreviations covered in this article) and SSA, to machine learning methods such as DT, SVM, and CNN. This diversity provides researchers with a wide range of tools to choose from based on their specific needs.

2. Performance metrics: Reported classification accuracies, such as approximately 99% achieved by CNN, provide quantitative measures of the effectiveness of certain methods. This information is crucial for researchers seeking high-performance models for microseismic waveform classification.

3. Data preprocessing challenges: Several studies have emphasized the importance of data preprocessing, including noise reduction and filtering. The time and effort required

for preprocessing, as mentioned in the study by Lin et al. [31], highlight the challenges in ensuring data quality before applying recognition models.

4. Need for larger datasets: Many studies have highlighted the importance of larger and more diverse training datasets for the robustness of model performance. For instance, Yi et al.'s [37] study emphasizes the need for extensive research to quantify the robustness of their method, indicating that dataset size could be a potential limiting factor.

5. Success of transfer learning: Dong et al.'s [51] demonstration of the success of transfer learning, where pretrained models like AlexNet and GoogLeNet outperform traditional SVM classifiers, suggests that leveraging existing knowledge in models could be a powerful strategy in microseismic event recognition.

6. Computational resource requirements: While deep learning methods are powerful, they demand a substantial amount of labeled data and computational resources. This underscores a practical consideration for researchers, especially those with limitations in data availability or computing infrastructure.

7. Appeal for comprehensive evaluation: Several studies call for larger-scale research to quantify the robustness and generalizability of their proposed methods. This emphasizes the need for a comprehensive evaluation framework to assess the practical applicability of developed models.

The verification of the stability and robustness of the model primarily includes (1) a stability assessment, which generally involves using techniques such as cross-validation, repeated experiments, or introducing noisy data to evaluate the consistency and stability of the model under different conditions and (2) robustness testing, which typically involves using adversarial samples to test the model's response to interference and perturbation. Additionally, introducing variations in input data, such as scaling, rotation, or translation, can be used to assess the model's robustness. Methods to enhance model robustness include using data augmentation techniques to expand the training dataset, applying regularization methods to reduce the risk of overfitting, and using ensemble learning methods to obtain more stable and robust predictions.

Checking the results of different studies and validate their reproducibility usually involves the following methods: (1) dataset examination: examine the dataset used in each study. Evaluate the quality, scale, and applicability of the dataset. Understand information such as the source of the dataset, data collection methods, and preprocessing steps to ensure the reliability and reproducibility of the dataset. (2) Method description: carefully read the detailed description of the methods used in each study. Understand details such as model architecture, hyperparameter settings, training strategies, etc. Ensure that the method description is sufficiently clear, allowing other researchers to replicate the experiments following the same steps. (3) Code sharing: check if there is code related to the study available for sharing. If so, attempt to reproduce the author's experimental results. Running the same code with the same dataset and settings can validate the reproducibility of the results. (4) Result comparison: compare the results of different studies. Pay attention to the consistency and differences between them. Focus on changes in performance metrics (such as accuracy, recall, etc.) and how the model performs on different datasets or tasks. (5) External validation: attempt to validate the results of other studies using similar datasets and tasks. Through repeated experiments, the model's generalization ability and reliability can be assessed. (6) Collaborative communication: collaborate and communicate with other researchers to discuss their experimental results with the same methods and datasets. Through communication, the reproducibility of the results can be further verified, and potential issues can be addressed collaboratively.

In summary, the comprehensive overview provided by Table 1 offers valuable insights for researchers in the field of microseismic event waveform recognition and classification. For example, spectral analysis methods are suitable for extracting frequency domain features but may overlook temporal information. Statistical methods provide rich data distribution information when dealing with microseismic waveforms, but they may face challenges in classifying complex waveforms. Traditional machine learning methods, such

as logistic regression, perform well in specific contexts but require manual feature engineering. In contrast, deep learning methods possess powerful automatic feature learning capabilities, enabling them to adapt to diverse waveforms. However, deep learning methods require a substantial amount of labeled data and computational resources. Researchers can leverage these findings to make informed decisions about which methods are best suited to their specific research goals.

The analyses have yielded a comprehensive understanding of the suitability of various methods in the domain of microseismic event waveform recognition and classification. These analyses effectively inform our choice of research methodology and shed light on the existing challenges and potential research avenues within this field. Consequently, this literature review serves as a pivotal resource, furnishing essential background information and invaluable guidance for researchers engaged in microseismic event waveform recognition and classification.

### 2.3. Identification Methods

This section discusses a range of methods used for microseismic event waveform identification, including traditional statistical methods, spectral analysis methods, and more advanced deep learning methods.

### 2.3.1. Statistical Analysis and Spectral Analysis

Statistical analysis and spectral analysis are traditional methods for microseismic event identification [52]. These methods aim to extract meaningful information by quantitatively analyzing specific attributes in microseismic event waveform data. In statistical methods, researchers typically focus on the statistical distribution, mean, variance, and other statistical characteristics of the signals to differentiate between different types of microseismic events. However, spectral analysis methods emphasize the frequency characteristics of the signals, such as spectral density and power spectral density, to distinguish the frequency domain features of microseismic events [53]. Although these techniques are simple and intuitive, they frequently require careful selection and feature extraction, which can be cumbersome when dealing with complicated and diverse waveforms.

Statistical analysis plays an important role in the recognition and classification of microseismic waveforms. Common statistical analysis methods include (1) mean and standard deviation: calculating the average and standard deviation of microseismic signals provides information about the overall level and variability of the signals; (2) correlation analysis: by calculating the correlation coefficient or correlation matrix between microseismic signals, the mutual relationship between the signals can be revealed, helping to determine their similarity or correlation; (3) probability distribution analysis: analyzing the probability distribution of microseismic signals, such as normal distribution, exponential distribution, etc., can provide insights into the distribution characteristics within different ranges, facilitating classification and discrimination; (4) statistical feature extraction: extracting statistical features of microseismic signals, such as kurtosis, skewness, energy, etc., can capture specific statistical patterns of the signals, assisting in identification and classification; (5) hypothesis testing: utilizing statistical hypothesis testing methods like *t*-test, analysis of variance, etc., to compare microseismic signals of different categories and examine if significant differences exist between them; and (6) cluster analysis: applying cluster analysis methods to group microseismic signals based on their features, thereby achieving waveform classification and recognition. Statistical analysis methods provide an overall description and summary of microseismic signal data, aiding in understanding the distribution patterns and features of the signals and providing a basis for subsequent recognition and classification. Furthermore, combining statistical methods with other techniques like machine learning and deep learning can further enhance the accuracy and reliability of waveform recognition and classification.

On the other hand, spectrum analysis reveals the frequency characteristics of signals by transforming them into the frequency domain. It involves key concepts and methods

such as (1) Fourier transform: Fourier transform is the foundation of spectrum analysis, converting signals from the time domain to the frequency domain, representing them as functions of frequency and amplitude. (2) Power spectral density: power spectral density describes the energy distribution of a signal across different frequencies, which can be obtained through the Fourier transform. It displays the intensity or energy distribution of the signal in different frequency bands, helping to identify the frequency components of the signal. (3) Fast Fourier transform (FFT): FFT is commonly used for efficient computation of signal spectra. It allows for the fast calculation of the frequency spectrum of discrete signals, improving computational efficiency. (4) Spectrogram: a spectrogram is a visual representation of the signal's frequency spectrum, with frequency on the horizontal axis and amplitude or energy on the vertical axis. It provides an intuitive display of the frequency components and energy distribution of the signal, aiding in observing spectral features. (5) Band energy features: by dividing the spectrum into different frequency bands and calculating the energy or power in each band, band energy features can be extracted. These features reflect the strength or energy distribution of the signal in different frequency ranges, facilitating the identification and classification of different types of microseismic signals. Spectrum analysis methods reveal the frequency characteristics of microseismic signals, helping to identify different frequency components and energy distributions, thus enabling waveform identification. It has widespread applications in microseismic monitoring, seismology, structural health monitoring, and other fields.

The processing workflow involved in statistical and spectral analysis methods is clearly depicted in Figure 4. Manual recognition methods typically rely on the expertise and intuition of domain experts to judge the event type by manually selecting and extracting waveform features. Traditional machine learning methods automate this process by training models to learn the features of different event types and then classify new waveform data. These methods perform well in some cases but still depend on manually designed features, which may have limitations in the classification and recognition of complex waveforms. In contrast, deep learning methods, as emerging technologies, possess powerful automatic feature learning capabilities and can directly extract features from raw waveform data, thus having greater potential in microseismic event recognition.

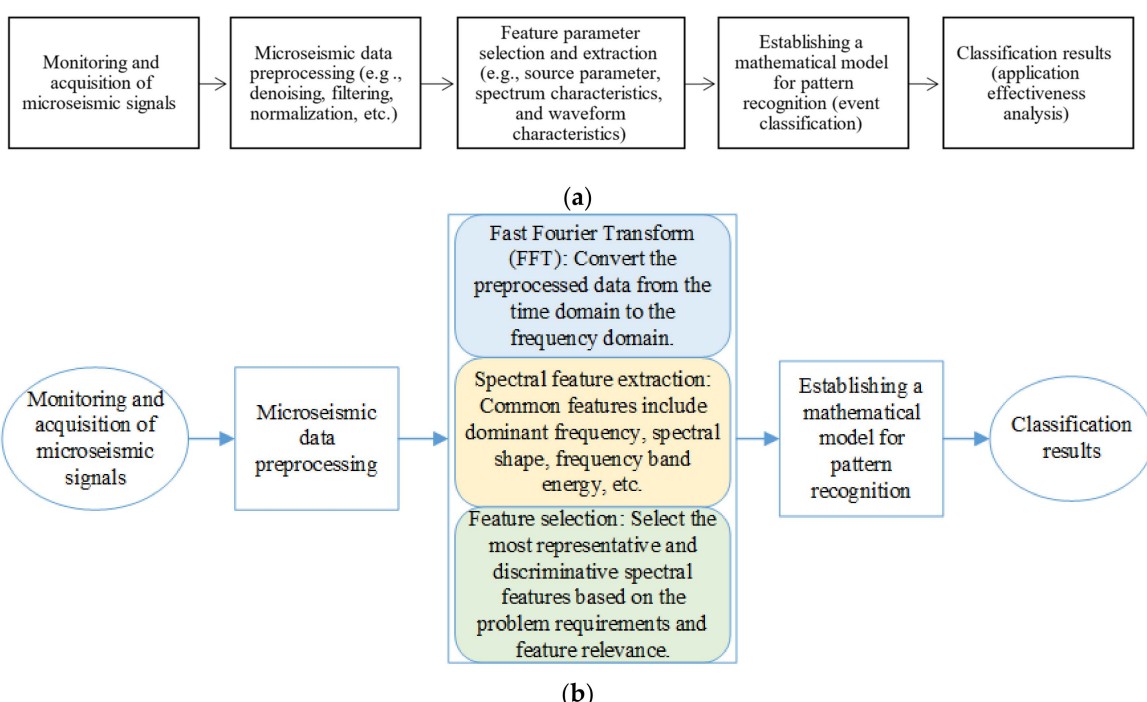

**Figure 4.** Workflow for recognizing different microseismic events using statistical and spectral analyses. (**a**) Statistical method (**b**) Spectral method.

### 2.3.2. Deep Learning Methods

Deep learning has emerged as a powerful tool in the field of microseismic event waveform image recognition [17]. Figure 5 illustrates the workflow for image recognition tasks using both traditional machine learning methods and deep learning techniques. In particular, CNNs are employed as the deep learning model, where multiple convolutional layers and pooling layers are stacked to extract local and global features from waveform data. These features capture important information such as the shape, frequency, and temporal characteristics of the waveforms. Subsequently, the extracted features are fed into fully connected layers for the purpose of microseismic event recognition and classification.

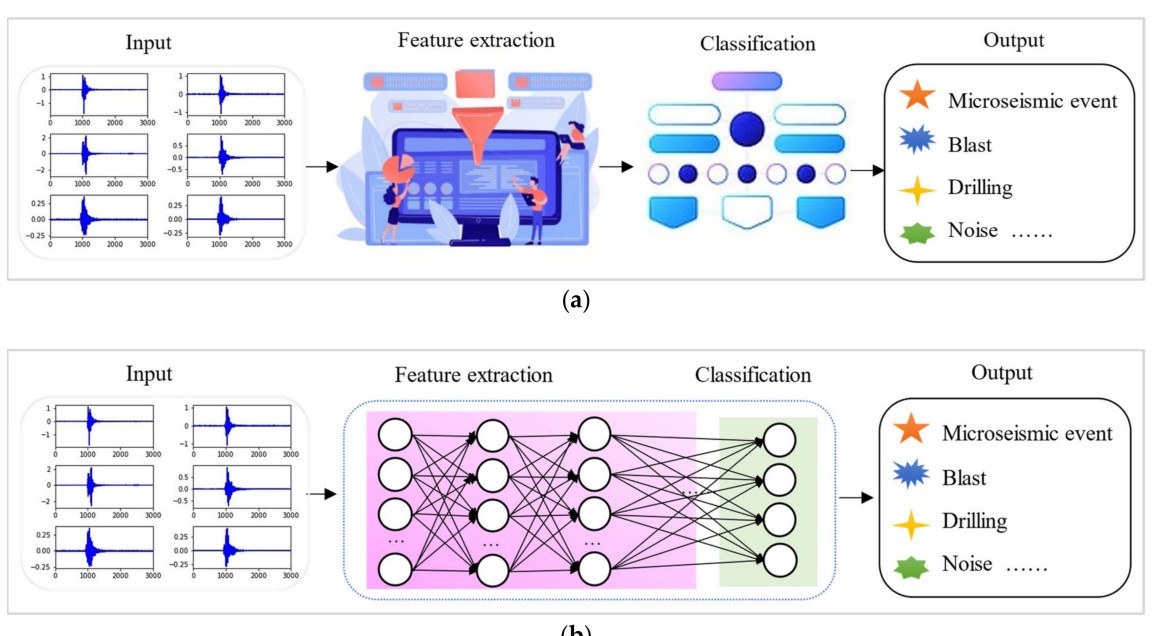

(a)

(b)

**Figure 5.** Image detection and classification using various machine learning models. (**a**) Traditional machine learning method (**b**) Deep learning.

Traditional machine learning methods have the following characteristics: (1) they require manual feature engineering, which involves selecting and designing relevant features for the problem at hand; (2) they have lower data requirements and are prone to overfitting when dealing with small datasets; (3) they offer faster computation speed and relatively simpler training processes; (4) in cases in which feature expression capability is limited, traditional machine learning methods may struggle to capture complex nonlinear relationships.

In contrast, deep learning methods exhibit distinct advantages in microseismic event image recognition. Firstly, they can automatically learn complex patterns and features from waveform data without the need for manual feature engineering. Secondly, deep learning models can hierarchically abstract features by extracting low-level local characteristics and high-level global features, enabling a more accurate understanding and representation of semantic information in microseismic event waveforms. Thirdly, deep learning benefits from large-scale dataset training, providing enhanced accuracy and generalization capabilities when dealing with noise and variability in microseismic event recognition tasks. However, it should be noted that deep learning methods may require more computational resources and time due to the training involved with multiple layers of neural networks.

Deep learning technology has shown promising potential for microseismic event image detection. It is not only capable of handling complex and diverse waveform data but also has the ability to learn features automatically, which brings great potential to the field of microseismic monitoring. This presents opportunities for achieving more precise

recognition and categorization of distinct types of microseismic events, contributing to improving security in subterranean engineering and mining operations.

## 3. Results and Discussion

In this section, we evaluate and discuss the major research achievements in using deep learning methods for microseismic event waveform image detection and classification. To comprehensively assess their effectiveness, we will explore the strengths, weaknesses, opportunities, and threats of existing research methods from four perspectives, known as SWOT analysis.

### 3.1. Classification of Methods

Based on existing studies, we further summarized the methods used in the field of microseismic event waveform recognition and classification. Specifically, we categorized these methods into seven categories: time domain analysis methods, frequency domain analysis methods, statistical feature analysis methods, machine learning methods, deep learning methods, cluster analysis methods, and hybrid methods. The explanations and analysis regarding the advantages and limitations of different types of methods are presented in Table 2, aiming to help readers better understand the characteristics and applicability of these methods.

**Table 2.** Classification and comparison of commonly used methods for microseismic signal identification and classification.

| Method | Description | Advantage | Disadvantage |
|---|---|---|---|
| Time domain analysis [54] | Classification based on time domain features such as the amplitude and duration of the waveform. | Simple and intuitive | Limited information richness, cannot capture complex patterns |
| Frequency domain analysis [55] | Extracts frequency features using spectrum analysis, such as dominant frequency and bandwidth, for classification. | Captures frequency-related information | Ignores time domain information, not suitable for non-stationary signals |
| Statistical feature analysis [56] | Extracts statistical features like mean, variance, skewness, etc., to describe waveform characteristics. | Simple computation, low resource consumption | Limited information richness, difficulty in capturing complex patterns |
| Machine learning methods [18] | Includes SVM, Random Forests, etc., mapping waveform features to a feature space for classification. | Suitable for complex relationships, handles high-dimensional data | Relies heavily on feature engineering, may require significant labeled data |
| Deep learning methods [57] | Uses deep learning models like CNNs for feature learning and classification, suitable for complex image information. | Automatically learns features, high adaptability | Requires large amounts of data, complex model parameter tuning, high computational resource consumption |
| Clustering analysis [58,59] | Clusters waveforms based on similarity, discovering natural groupings in the data. | Can identify unknown categories | Results are highly dependent on the choice of distance measure, clustering algorithm, etc. |
| Hybrid methods [43] | Combines multiple methods, e.g., integrating machine learning with deep learning, or combining time–frequency analysis with statistical feature analysis. | Considers multiple aspects, improves classification accuracy | Requires more data and computational resources, increased algorithm complexity |

Time domain analysis focuses on the variations in time and waveform characteristics of a signal. By observing the amplitude, duration, periodicity, and other characteristics of the signal, intuitive perceptions and time-related information about the signal can be obtained. This information is very useful for the identification and location of microseismic events. Common time domain analysis methods include waveform plots, envelope analysis, autocorrelation functions, and cross-correlation functions. Time domain analysis methods are suitable for detecting transient events, impulse signals, and time correlations. On the other hand, frequency domain analysis is concerned with the components and

spectral characteristics of a signal in terms of frequency. By converting the signal to a frequency domain representation, the frequency distribution, frequency components, and corresponding energy distribution of the signal can be analyzed. Common frequency domain analysis methods include Fourier transform, power spectral density, and spectrogram. Frequency domain analysis methods are suitable for analyzing the frequency characteristics of signals, the interrelationships between frequency components, etc. Therefore, time domain and frequency domain analyses are two commonly used methods in microseismic signal processing. Time domain analysis focuses on the temporal variations and waveform characteristics of the signal, providing intuitive time-dependent information, whereas frequency domain analysis focuses on the frequency components and energy distributions of the signal, revealing the frequency characteristics of the signal. These two methods complement each other and together provide a comprehensive description and understanding of the signal, which helps to identify microseismic events and study their frequency characteristics and time correlation.

Statistical feature analysis involves extracting and calculating various statistical features from a signal. These statistical features include mean, standard deviation, peak, energy, etc., and can be used to characterize the signal as a whole by analyzing its statistical properties. Statistical feature analysis methods are usually based on mathematical–statistical theory, can be computed quickly, and do not require a large number of training samples. These features can be used to construct rule- or threshold-based classification models. Machine learning methods are a data-driven approach that learns signal features and performs classification using training models. Common machine learning methods include SVM, RF, and naive Bayes. These methods typically require manual feature design and use the extracted features to train classification models. Machine learning methods are suitable for moderate-sized datasets and relatively simple problems, but they may have limitations in dealing with complex signal patterns. Deep learning methods, on the other hand, are based on neural networks and can automatically learn features and patterns through multilayered networks for classification. The core of deep learning methods is ANN [60], such as CNNs and RNNs. These methods optimize network weights using large-scale training data and backpropagation algorithms to automatically extract complex signal features and perform advanced classification. Deep learning methods excel in handling large-scale datasets and complex problems.

The difference between machine learning and deep learning methods lies in the former requiring manual design of feature extraction processes and using the extracted features to train classification models, whereas the latter automatically learns to signal features and patterns through multilayer neural networks without the need for manual feature definition. Deep learning methods typically require more training data and computational resources, but they can handle more complex signal patterns and have stronger generalization capabilities. Therefore, in microseismic signal recognition and classification, machine learning methods are suitable for moderate-sized datasets and relatively simple problems, whereas deep learning methods are suitable for large-scale datasets and complex problems. Researchers can choose the appropriate method based on the dataset size, problem complexity, and available resources.

Clustering analysis is an unsupervised learning method that divides data samples into different groups or clusters, grouping similar samples together. It focuses on discovering inherent patterns and structures in the data without the need for prior labels or category information. Common clustering algorithms include k-means, hierarchical clustering, and DBSCAN. Clustering analysis can help identify similar patterns and clusters within microseismic signals for further investigation and classification. Hybrid methods combine multiple techniques or methods to leverage their respective strengths. In microseismic signal recognition and classification, hybrid methods often integrate multiple techniques or methods, such as combining clustering analysis with machine learning methods or combining clustering analysis with statistical feature analysis. The goal of hybrid methods is

to obtain more comprehensive features from multiple perspectives and fuse the advantages of different methods to improve recognition and classification performance.

By categorizing and evaluating these methods, we found that each method has its advantages and limitations. Researchers can choose the appropriate method or combine multiple methods according to their specific research objectives and data characteristics to achieve accurate and efficient identification and classification.

### 3.2. SWOT Analysis of Deep Learning

Here is a concise overview of the SWOT analysis results for using deep learning methods in the research of microseismic event waveform image recognition and classification:

- Strengths

(1) Sensitivity to subtle changes: Deep learning excels at capturing intricate patterns and subtle variations in microseismic waveform images, enabling precise event recognition and classification. For example, Huang et al. [61] successfully identified and located microseismic events using CNN.

(2) Feature extraction: Deep learning techniques effectively extract complex features from waveform images, thereby enhancing the differentiation of various microseismic events. For instance, Li et al. [15] achieved high accuracy in microseismic waveform classification using DCNN.

(3) Scalability: Deep learning's capacity to handle large datasets aligns well with the data-intensive nature of microseismic monitoring, thus leading to improved accuracy. Wang et al. [47] applied deep learning methods to process a large volume of microseismic waveform data, resulting in good classification results.

- Weaknesses

(1) Limited labeled data: The scarcity of labeled microseismic waveform data can hinder model training and validation, potentially limiting the algorithm's performance. For example, Wang et al. [47] noted a rapid decline in classification accuracy when the number of training samples was less than 512.

(2) Resource demands: Complex deep learning models may require substantial computational resources, which poses challenges in resource-constrained environments. For instance, Dong et al. [51] mentioned that training deep convolutional neural networks requires more time and computational resources than traditional classification methods.

(3) Expertise requirement: Implementing and fine-tuning deep learning models requires expertise in both microseismic domain knowledge and machine learning. For example, Bi et al. [38] demonstrated that the parameter tuning and debugging of deep learning models require machine learning expertise, while the interpretability of results also relies on knowledge in the microseismic domain.

- Opportunities

(1) Continuous advancements: The ongoing development of deep learning methodologies promises enhanced accuracy and performance in microseismic event recognition and classification. For example, Dong et al. [51] demonstrated the successful application of pretrained models in microseismic event recognition, indicating that advancements in deep learning methods bring better opportunities in this field.

(2) Versatile applications: The adaptability of deep learning methods opens doors to broader applications beyond microseismic monitoring, offering potential insights into related domains. For instance, Mousavi et al. [4] explored the possibility of applying deep learning methods to seismic data processing and achieved promising results.

- Threats

(1) Data privacy concerns: The sharing and utilization of sensitive microseismic data could be constrained by privacy and security considerations.

(2) Noise interference: The presence of noise within microseismic waveform images might challenge the robustness and effectiveness of deep learning models. For instance,

Xu et al. [62] pointed out that noise interference may lead to performance degradation in waveform classification tasks.

This SWOT analysis underlines the potential of deep learning techniques in microseismic event waveform image recognition and classification, while also acknowledging challenges related to data availability, resource demands, and the need for domain expertise.

*3.3. Comparison of Deep Learning Models*

Deep learning methods play a crucial role in microseismic data analysis. By automatically learning and extracting features from waveform data, these methods provide powerful tools for seismologists and engineers to understand subsurface structures and seismic activities. However, having a clear understanding of the advantages and disadvantages of different deep learning methods is essential when choosing the appropriate approach. Table 3 provides descriptions, advantages, and disadvantages analysis for several common deep learning methods. From CNNs that capture spatial features to LSTM networks that handle temporal dependencies, each method has its own characteristics and application scope. Additionally, transfer learning, attention mechanisms, and ensemble methods have been introduced to further enhance the classification performance and information extraction capabilities of microseismic data. A comprehensive evaluation of these deep learning methods can assist researchers in selecting the most suitable approach for their research objectives and drive the development and innovation in the field of microseismic data analysis.

**Table 3.** A summary and comparison of commonly used deep learning methods for microseismic event waveform recognition and classification.

| Deep Learning Method | Description | Advantage | Disadvantage |
|---|---|---|---|
| Convolutional neural networks (CNNs) | Utilizes convolutional layers to automatically learn hierarchical features from raw waveform images. | Captures spatial features, strong in image data | Requires large labeled datasets, complex model architectures |
| Recurrent neural networks (RNNs) | Suited for sequential data, processes waveforms in a time series manner, capturing temporal patterns. | Captures temporal dependencies, adaptable | Vulnerable to vanishing/exploding gradient problem, less suitable for complex spatial patterns |
| Long short-term memory (LSTM) | A type of RNN designed to avoid long-term dependency issues that effectively captures long-range temporal information. | Handles long-range dependencies, suitable for time series | More complex to implement, and requires careful tuning and training |
| Gated recurrent units (GRUs) | Another type of RNN that balances complexity and performance and is similar to LSTM but with fewer parameters. | Efficient memory usage, adaptable | May struggle with very long sequences, less expressive than LSTM |
| Transfer learning | Utilizes pretrained models on large datasets to extract general features and then fine-tunes for specific microseismic data. | Requires less labeled data, faster convergence | Depending on the availability of relevant pretrained models, potential domain gap issues |
| Attention mechanisms | Enhances feature extraction by assigning different weights to different parts of the input waveform, focusing on important details. | Improves information retention, interpretable | May introduce additional model complexity, require more computational resources |
| Ensemble methods | Combines multiple deep learning models to improve classification performance and reduce individual model bias. | Increases robustness, balances model shortcomings | Requires additional computational resources, may be complex to implement |

Taking CNNs as an example, we have summarized the advantages and disadvantages of commonly used classical image classification models (AlexNet, GoogLeNet, ResNet) and the standard CNN model. Specific details can be found in Table 4. This analysis

complements the findings in Table 3 and provides a more in-depth discussion. Overall, these classical image classification models have all solved the challenges in deep convolutional neural networks to varying degrees, and have achieved remarkable results in image classification tasks. Selecting a suitable model requires weighing the advantages and disadvantages according to the requirements of the task and the available resources.

**Table 4.** Comparison of classical image classification models.

| Model | Advantages | Disadvantages |
|---|---|---|
| Standard CNN | (1) Intuitive and easy to understand. (2) Suitable for small-scale image classification tasks. (3) Faster training speed compared to deeper models. | (1) Misses advanced model optimizations. (2) May struggle with complex tasks or large datasets. |
| AlexNet | (1) First successful deep learning model on ImageNet. (2) Introduced ReLU activation for faster convergence. (3) Used Dropout to mitigate overfitting. | (1) Large number of parameters, prone to overfitting. (2) Deeper architecture leads to longer training times. |
| GoogLeNet | (1) Utilized inception modules to reduce parameters. (2) Achieved good performance with fewer layers. (3) Balanced depth and performance. | (1) Complex architecture, hampers module interpretation. (2) Longer training time due to increased complexity. |
| ResNet | (1) Introduced residual blocks to address gradient vanishing. (2) Enabled extremely deep networks with good results. (3) Can increase depth without degradation in performance. | (1) Increased complexity demands more computational resources. (2) Potential overfitting on small datasets. |

### 3.4. Opportunities and Challenges

Deep learning models face numerous opportunities and challenges in their future development, which will directly impact their applications and performance in the field of microseismic monitoring. Here are some key opportunities and challenges for deep learning models in their future development:

- Opportunities

(1) Availability of large-scale datasets: As monitoring technologies advance, the volume and diversity of seismic waveform and microseismic event data will increase. This will provide deep learning models with more training data, enhancing their performance and generalization capabilities in the recognition and classification of microseismic event waveform images.

(2) Advancements in computational power: Rapid developments in hardware technologies, particularly graphics processing units (GPUs) and specialized deep learning chips like TPUs, will significantly boost the training speed and efficiency of deep learning models. This will enable the handling of large-scale data and more complex models.

(3) Transfer learning and pretrained models: Pretrained models and transfer learning methods have achieved tremendous success in various fields. These techniques allow models trained in one domain to be transferred to another, thereby reducing data requirements and training time. For microseismic monitoring, this means that existing deep learning models can be more easily applied and fine-tuned for specific problems.

(4) Automation and tools: Evolving automation tools and platforms, such as AutoML and deep learning frameworks, enable a broader audience to utilize deep learning techniques without delving deep into the underlying details. This will drive the proliferation and wider application of deep learning.

(5) Interdisciplinary Collaboration: In navigating the intricacies of integrating machine learning with Earth science, several potential interdisciplinary collaboration models emerge, bolstered by successful partnerships in analogous domains: (a) joint research projects: Forming collaborative research teams comprising both machine learning experts and Earth science specialists to jointly undertake specific projects. For instance, in earthquake prediction, machine learning experts collaborate with seismologists, employing machine learning algorithms to analyze and forecast seismic data, thereby enhancing earth-

quake early warning capabilities. (b) Data sharing and integration: facilitating collaboration through data-sharing platforms, enabling machine learning experts to access real-world Earth science data. Earth science experts, in turn, leverage machine learning techniques to decipher and extract latent information from these datasets. In climate change research, for example, machine learning experts utilize extensive meteorological observation data to improve and optimize climate models using machine learning algorithms. (c) Institutional collaboration and resource sharing: establishing collaborative relationships between institutions for the shared use of laboratory equipment, datasets, and research resources. In geological exploration and resource development, collaboration between machine learning experts and geologists improves exploration efficiency and accuracy by sharing geological data and machine learning algorithms. (d) Interdisciplinary research centers: creating specialized centers that bring together experts from machine learning and Earth science domains for cutting-edge research. This model facilitates in-depth interdisciplinary collaboration, offering a robust platform for addressing complex problems.

In short, the integration of machine learning with Earth science in microseismic monitoring is driven by factors such as the availability of large-scale datasets, advancements in computational power, transfer learning and pretrained models, automation tools, and interdisciplinary collaboration. These factors enhance model performance, enable efficient training, reduce data requirements, facilitate wider application, and foster collaborative research. This integration has great potential for advancing microseismic monitoring capabilities.

- Challenges

(1) Real-time and accuracy: Real-time identification and classification of microseismic events pose significant challenges, especially when dealing with large volumes of data. Models must make accurate decisions within extremely short timeframes, which requires improvements in model design and computational efficiency. Balancing real-time requirements with model complexity and computational demands is crucial.

(2) Generalization of models to novel events: Generalizing models to novel event types is a major challenge because models often have limited exposure to certain event categories during training. This necessitates further research into transfer learning and data augmentation methods to adapt to evolving monitoring environments.

(3) Model interpretability: In the domain of microseismic events, the interpretability of models is essential for ensuring decision transparency and credibility. Currently, deep learning models are often considered "black-box" models, making it difficult to explain their decision-making processes. Therefore, future research should focus on enhancing the interpretability of deep learning models.

(4) Imbalanced data: Microseismic event data frequently suffer from class imbalance, where some event categories have fewer samples. This can lead to performance degradation in minority classes and requires targeted solutions such as oversampling or generative adversarial networks (GANs).

(5) Sustainability and resource efficiency: Deep learning models typically require substantial computational resources, which poses sustainability and resource-related challenges. Therefore, research efforts should focus on improving model resource efficiency, including the development of more energy-efficient hardware and algorithms.

Based on the aforementioned opportunities and challenges, the field of microseismic monitoring can further promote the development and application of deep learning through the following aspects: (1) data processing and model optimization: to address the limited labeled data issues, techniques such as semi-supervised learning, active learning, and transfer learning can be explored to reduce the reliance on a large amount of labeled data. This will help improve model performance and generalization capabilities. To mitigate noise interference, research should focus on noise removal techniques, waveform restoration methods, and data augmentation strategies to enhance the robustness of deep learning models against noise. (2) Model interpretability and credibility: in the development of deep learning models, emphasis should be placed on improving their interpretability.

Researchers can explore the use of interpretable deep learning architectures, introduce attention mechanisms, or employ explainable models to elucidate the decision-making process. Additionally, establishing uncertainty estimation methods for models is crucial as it helps evaluate the confidence of model outputs and provides more reliable decision-making support. (3) Heterogeneous dataset and class imbalance issues: strategies such as oversampling, undersampling, and GANs can be employed to address these challenges. These methods assist in tackling the problem of insufficient samples in minority classes and improve model performance across all categories. (4) Model deployment and real-time capabilities: to tackle the challenges of real-time requirements and accuracy, research should focus on optimizing the inference speed and computational efficiency of deep learning models. Techniques like model compression, quantization, and hardware acceleration can be explored to meet real-time monitoring needs. (5) Interdisciplinary collaboration and knowledge integration: cross-disciplinary collaboration is essential when facing challenges and opportunities. Collaborating with experts from computer science, seismology, geology, physics, and other fields facilitates the fusion and exchange of knowledge, thus contributing to the advancement of the microseismic monitoring field.

In summary, by focusing on data processing and model optimization, model interpretability and credibility, heterogeneous datasets and class imbalance issues, model deployment and real-time capabilities, and fostering interdisciplinary collaboration and knowledge integration, the field of microseismic monitoring can provide more accurate, efficient, and reliable solutions. This will drive advancements in science and technology within this domain.

## 4. Conclusions

This study aims to explore the application of machine learning techniques in the field of microseismic event waveform image recognition and classification, as well as their potential and limitations. Through a comprehensive analysis and discussion of the relevant literature, we have drawn the following conclusions:

1. Early spectrum analysis methods in microseismic event recognition relied heavily on domain knowledge and expertise, requiring experts to engage in time-consuming manual analysis and interpretation. Another common statistical analysis method classified microseismic events based on statistical features, but it required manual definition and selection of appropriate features, as well as manual extraction and analysis. Moreover, traditional manual recognition methods suffer from heavy workloads and low efficiency.

2. In contrast, adopting machine learning methods such as deep learning can improve recognition efficiency and accuracy by automatically learning features and pattern recognition, thereby reducing the burden of manual involvement. These methods are more effective in handling large-scale datasets and enhancing the automation of microseismic event recognition.

3. Combining machine learning algorithms with computer vision techniques provides a direct solution for analyzing microseismic event waveform images, with the ability to handle large-scale datasets and enhance the precision and speed of event recognition and classification.

4. However, in the field of microseismic event recognition, deep learning algorithms face challenges such as insufficient annotated data, high computational resource requirements, and the need to improve generalization performance. Addressing these difficulties requires interdisciplinary collaboration and ongoing research efforts.

5. Future directions include several aspects: firstly, establishing strategies to improve the robustness of models, thereby enhancing the stability and reliability of microseismic event waveform recognition and classification models. This can be achieved through the use of more complex network structures, introducing regularization techniques, or integrating multiple models. Secondly, creating diverse datasets is crucial. These datasets should contain abundant samples of microseismic events, covering various geological environments and working conditions. Training and evaluating models using diversified datasets can im-

prove their generalization ability and adaptability. Thirdly, data augmentation techniques can effectively increase the diversity and quantity of training data, thereby enhancing model performance. For example, data augmentation can be achieved through operations such as rotation, scaling, and translation to expand the dataset, allowing the model to better adapt to variations and differences among microseismic events. Furthermore, exploring transfer learning and self-supervised learning is a promising research direction in microseismic event recognition using deep learning. Transfer learning can leverage existing models and knowledge to quickly adapt to new microseismic event recognition tasks, thereby improving model performance and generalization ability. Self-supervised learning can use appropriate methods to generate labels automatically, utilizing unsupervised data for training and further enhancing model performance. Through continuous exploration and improvement, deep learning will provide more reliable safety guarantees for mining engineering in the field of microseismic monitoring.

In conclusion, deep learning approaches provide effective instruments for microseismic event identification, with the potential to increase the accuracy and efficiency of microseismic event categorization, as well as provide more dependable security guarantees for underground engineering and mining activities. However, further study and effort are required to solve the current problems and ensure the long-term implementation of deep learning approaches in microseismic monitoring.

**Author Contributions:** Conceptualization, A.Y.D.; methodology, H.S.; validation, A.Y.D.; formal analysis, H.S.; investigation, H.S.; resources, A.Y.D.; data curation, H.S.; writing—original draft preparation, H.S.; writing—review and editing, A.Y.D.; visualization, H.S.; supervision, A.Y.D.; project administration, A.Y.D.; funding acquisition, A.Y.D. All authors have read and agreed to the published version of the manuscript.

**Funding:** This research received no external funding.

**Institutional Review Board Statement:** Not applicable.

**Informed Consent Statement:** Not applicable.

**Data Availability Statement:** No new data were created or analyzed in this study. Data sharing is not applicable to this article.

**Acknowledgments:** The authors sincerely appreciate the guidance and comments provided by the handling editor and anonymous reviewers for this research. Additionally, the authors would like to express gratitude to all the participants and relevant institutions involved in this study, as their participation and contributions have been essential to the outcome of this research.

**Conflicts of Interest:** The authors declare no conflict of interest.

## Appendix A

**Table A1.** Glossary of abbreviations.

| Abbreviation | Full Name | Abbreviation | Full Name |
|---|---|---|---|
| CNN | Convolutional Neural Network | XTF-CNN | Time–Frequency Convolutional Neural Network |
| SWOT | Strengths, Weaknesses, Opportunities, and Threats | LSTM | Long Short-Term Memory |
| IoT | Internet of Things | RNN | Recurrent Neural Network |
| AI | Artificial Intelligence | FCN | Fully Convolutional Neural Networks |
| MS | Microseismic Signal | ResNet | Residual Network |
| MMS | Microseismic Monitoring System | PSO | Particle Swarm Optimization |
| DT | Decision Tree | SSA | Singular Spectrum Analysis |
| SVM | Support Vector Machine | BP | Back Propagation |
| DCNN | Deep Convolutional Neural Network | LDA | Linear Discriminant Analysis |
| SPP | Spatial Pyramid Pooling | WSD | Wavelet Scattering Decomposition |

**Table A1.** *Cont.*

| Abbreviation | Full Name | Abbreviation | Full Name |
|---|---|---|---|
| DAS | Distributed Acoustic Sensing | VGG | Visual Geometry Group |
| STA/LTA | Short-Term Average/Long-Term Average | ECNN | Enhanced Convolutional Natural Network |
| RF | Random Forests | GAN | Generative Adversarial Networks |
| KNN | K-Nearest Neighbors | ACGAN | Auxiliary Classifier Gan |
| EEMD | Ensemble Empirical Mode Decomposition | WPD | Wavelet Packet Decomposition |
| SVD | Singular Value Decomposition | STFT | Short-Time Fourier Transform |
| ELM | Extreme Learning Machine | NB | Naive Bayes |
| DBN | Deep Belief Network | ANN | Artificial Neural Networks |
| CapsNet | Capsule Network | DBSCAN | Density-Based Spatial Clustering of Applications with Noise |
| PCA | Principal Component Analysis | GRU | Gated Recurrent Unit |
| CEEMDAN_SE | Complete Ensemble Empirical Mode Decomposition with Adaptive Noise Sample Entropy | GPU | Graphics Processing Unit |

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
