# Peer review of "Microseismic Monitoring Signal Waveform Recognition and Classification: Review of Contemporary Techniques"

_applsci, doi:10.3390/app132312739_

Round 1

Reviewer 1 Report

Comments and Suggestions for Authors

The review paper on machine learning techniques for microseismic event waveform image recognition and classification appears to be comprehensive and methodical in its approach. Here are some comments and suggestions for improvement:

  1. Comprehensiveness of Literature Review: The paper has referenced a range of sources from credible databases and categorized studies according to their significance, which is commendable. However, it should ensure that the literature encompasses the full scope of the field, including any conflicting or negative results, to present a balanced view.

    Improvement Suggestion: Verify that the review includes diverse perspectives and not just those that support a particular viewpoint or technology. This could involve actively seeking out literature that presents challenges or limitations to the current consensus.

  2. Methodological Analysis: The paper offers a classification of methods and a detailed discussion of their respective strengths and weaknesses, which is crucial for a review paper. However, the analysis should also critically assess the validity of the methodologies used in the primary studies and their results.

    Improvement Suggestion: Incorporate a discussion on the reliability of the different machine learning methods, possibly by examining how different studies validate their results and the reproducibility of these results.

  3. SWOT Analysis: The SWOT analysis provides a strategic insight into the application of deep learning methods in microseismic event classification. However, it should be backed by specific examples or case studies from the literature to illustrate these points concretely.

    Improvement Suggestion: Include real-world examples or scenarios where the strengths and opportunities have been leveraged, or where the weaknesses and threats have manifested, to give the SWOT analysis more depth and practical relevance.

  4. Discussion of Challenges and Opportunities: The paper correctly identifies challenges such as the need for labeled data and computational resources, as well as opportunities like the advancement of computational power and learning technologies. However, these could be expanded to include how the field might evolve in response to these challenges and opportunities.

    Improvement Suggestion: The discussion could be enhanced by proposing potential solutions to the identified challenges, such as the development of semi-supervised learning techniques that require less labeled data or the exploration of more efficient deep learning architectures.

  5. Future Research Directions: The conclusion provides a forward-looking perspective, suggesting avenues like data augmentation and transfer learning. Yet, it would benefit from more specificity regarding how these suggestions could be implemented.

    Improvement Suggestion: Detail a roadmap or framework for future research, providing clearer steps on how to address the current limitations and what metrics or benchmarks should be used to measure progress.

  6. Interdisciplinary Collaboration: While the need for interdisciplinary collaboration is acknowledged, the paper could offer more concrete suggestions on how this can be achieved, considering the complexity of combining machine learning with geoscience.

    Improvement Suggestion: Outline potential interdisciplinary collaborative models and provide examples of existing partnerships that have been successful in solving complex problems in similar domains.

In essence, the review paper seems to be on the right track with a structured analysis of the current state of machine learning applications in microseismic event classification. However, by incorporating more critical analysis, real-world examples, concrete future steps, and collaborative frameworks, it could provide a more actionable and insightful resource for researchers and practitioners in the field.

Author Response

We sincerely appreciate the valuable feedback and suggestions provided by the reviewer. We have carefully read your comments and have responded to each of them individually. Please refer to the attached document for our detailed responses.

Reviewer 2 Report

Comments and Suggestions for Authors

The article provides a comprehensive review of various techniques Micro-seismic Monitoring Signal Waveform (MMS) Recognition and Classification. The paper is generally well-written and balanced. However, following can further improve the paper.

Title: Since study review various techniques for the said problem, why only machine learning has been mentioned in the title. In my opinion, title should cover a broader view since it is a review article. For instance:

"Microseismic Monitoring Signal Waveform Recognition and Classification: Review of Contemporary Techniques"

Abstract: There is need to expand the abstract. Do add SWOT outcomes in a brief in the abstract as well.

Introduction: The introduction need to address the problem in a broader way, not just smart mining. Add more potential applications. Moreover, it is good to add some statistics regarding the application areas.

By the end of introduction, kindly add study's layout. Like: 

The rest of the article is organized as follows: section 2 presents material and methods.....

Materials and Methods: Kindly use proper citations in the Table. For instance:

B. I. Lin et al., 2018 [30] should be "Lin et al., 2018"

Paper uses a lot of acronyms and abbreviations. It is good to add a table for them for a better readership. Depending on the journal's template, it could be in the appendix.

Followed by Table 1, add few concrete points as the takeaway. Currently, the paragraph after the table, presents a generic overview. Same applied to the subsequent section "Identification methods."

Moreover, if possible, kindly add the growth rate of ML/DL methods over the time in the said area of study in contrast to the traditional methods.

Results and Discussion: In Table 2 and Table 3 comparisons are made. DL methods are enlisted in Table 2 as well. Just if you can provide a justification. For instance, would it more useful that non-DL methods should be compared separately, and DL method separately and then contrast is drawn. Adding few lines would help reader get out of this confusion.

Conclusion: Though well-written, conclusion should be more elaborated. Currently, it is too brief. Do expand it by adding more to the points 1-4.

Comments on the Quality of English Language

The paper is generally well written and adequately composed. After addressing the comments, it can be accepted for publication.

Author Response

(The authors gave the same response as above.)

Round 2

Reviewer 1 Report

Comments and Suggestions for Authors

The authors have addressed all the comments given.

Reviewer 2 Report

Comments and Suggestions for Authors

All comments are addressed.